# Fusion-Based Neutron Generator Production of Tc-99m and Tc-101: A Prospective Avenue to Technetium Theranostics

**DOI:** 10.3390/ph14090875

**Published:** 2021-08-29

**Authors:** Edward J. Mausolf, Erik V. Johnstone, Natalia Mayordomo, David L. Williams, Eugene Yao Z. Guan, Charles K. Gary

**Affiliations:** 1Innovative Fuel Solutions LLC, North Las Vegas, NV 89031, USA; ejmausolf@gmail.com; 2Helmholtz-Zentrum Dresden–Rossendorf (HZDR), Institute of Resource Ecology, Bautzner Landstraße 400, 01328 Dresden, Germany; n.mayordomo-herranz@hzdr.de; 3Adelphi Technology, Inc., Redwood City, CA 94063, USA; David@adelphitech.com (D.L.W.); eguan@adelphitech.com (E.Y.Z.G.); cgary@adelphitech.com (C.K.G.)

**Keywords:** technetium-99m, molybdenum-99, neutron generator, fusion, medical isotopes, radiopharmaceuticals, diagnostic, theranostic, special nuclear material (SNM)

## Abstract

Presented are the results of ^99m^Tc and ^101^Tc production via neutron irradiation of natural isotopic molybdenum (Mo) with epithermal/resonance neutrons. Neutrons were produced using a deuterium-deuterium (D-D) neutron generator with an output of 2 × 10^10^ n/s. The separation of Tc from an irradiated source of bulk, low-specific activity (LSA) Mo on activated carbon (AC) was demonstrated. The yields of ^99m^Tc and ^101^Tc, together with their potential use in medical single-photon emission computed tomography (SPECT) procedures, have been evaluated from the perspective of commercial production, with a patient dose consisting of 740 MBq (20 mCi) of ^99m^Tc. The number of neutron generators to meet the annual 40,000,000 world-wide procedures is estimated for each imaging modality: ^99m^Tc versus ^101^Tc, D-D versus deuterium-tritium (D-T) neutron generator system outputs, and whether or not natural molybdenum or enriched targets are used for production. The financial implications for neutron generator production of these isotopes is also presented. The use of ^101^Tc as a diagnostic, therapeutic, and/or theranostic isotope for use in medical applications is proposed and compared to known commercial nuclear diagnostic and therapeutic isotopes.

## 1. Introduction

^99m^Tc (t_1/2_ = 6.007 h; IT 99.9%) is the single most used isotope for all nuclear diagnostic-imaging procedures—it accounts for a majority of the radiopharmaceutical market, and essentially encompasses the whole of the single-photon emission computed tomography (SPECT) modality in the low-energy field [1]. There are several production routes for generating ^99m^Tc [2,3], although the primary means of commercially derived ^99m^Tc is produced from ^99^Mo (t_1/2_ = 65.924 h; β^−^ 100%) via neutron-driven nuclear reactions, of which, are dependent upon and characterised by the chemical and isotopic composition of the target material. For example, the use of either natural or enriched Mo or enriched U (20–90% ^235^U) targets. Under these scenarios, the ^98^Mo or ^235^U targets are irradiated with a neutron source, such as a nuclear reactor, and form ^99^Mo via ^98^Mo(n,γ)^99^Mo or ^235^U (n, fission) ^99^Mo (~6% fission yield), respectively. The generated ^99^Mo being sent to a processor is isolated, refined, and loaded onto a column, where Tc retention is negligible, such as alumina (Al_2_O_3_) [4], and fixed in a portable generator for subsequent distribution. At the receiving end, whether it be a nuclear pharmacy or a hospital, the ^99m^Tc is eluted from the generator, compounded, and delivered to the point of use where it is administered to a patient. However, during this process a significant quantity of ^99^Mo/^99m^Tc produced in this manner is typically lost due to decay (~1% per hour); the drug delivered at the end-patient user routinely accounts for less than 10% of the initial activity isolated from the reactor or the irradiation source at end of bombardment (EOB). 

Because of the relatively short half-lives of both the parent (^99^Mo) and the daughter (^99m^Tc), the radioisotope pair must continually be produced and distributed, as a long-term supply cannot be stockpiled. It has been said that this constraint impacts the possible production and distribution modes and models, the quantity of waste generated during production [5], and critically, the cost burden of the drug due to unused radioisotope lost to decay; the supply side is recognised [1] as being sub-optimal. 

The supply chain has been interrupted several times in recent years leading to shortages of the drug(s) [6,7,8,9]. Over the last decade, efforts to integrate redundancy within the production side of the supply chain (coordinated scheduling, back-up producers, varying production loads, etc.) have been a major focus of the industry. Although, issues such as unplanned outages [10], accidents [11], and maintenance of the nuclear reactors have led to varying production loads. As the current infrastructure and its resources succumb to ageing, where key nuclear reactors will eventually be brought offline and decommissioned, it is probable that further logistical and technical issues will arise. 

Bottlenecks in the supply-chain can occur as a result of the small number of ^99^Mo processors [1]. Reported consequences have been the inflation of drug pricing [12] and a reduction of emerging competition into the market to help combat rising costs [13]. The issue is further complicated by a lack of funding and issues related to the integration and implementation of waste processing and long-term disposal into the full-cost recovery (FCR) model further complicate the issue [1] and had not been entirely unforeseen [14].

In this paper, experimental production of ^99^Mo/^99m^Tc and ^101^Mo/^101^Tc using a neutron generator without the use of uranium (U) or special nuclear material (SNM) is presented. The potential impact that neutron generator-produced ^99^Mo/^99m^Tc and ^101^Mo/^101^Tc can have on the future of both diagnostic and theranostic modalities is considered. Whereas ^99m^Tc is typically considered a purely diagnostic isotope [15], it is envisioned that ^101^Tc (t_1/2_ = 14.22 min) could be applied as an ephemeral theranostic agent; a beta emission (~487 keV, 90.3% [16]) for radiotherapeutics is complemented by a 307 keV (89.4%) gamma emission ideal for synchronised diagnostic scanning [17]. In addition, the ^99m^Tc/^101^Tc pair may be useful in dual imaging modalities [18], where the similar chemistry, but differing nuclear properties of the two isotopes could be exploited.

Neutron generator produced isotopes coupled with real-time use, may off-set to an extent (or minimises) fission product waste from uranium post-processing, increases more efficient use of ^99m^Tc, and potentially coincides with the requirements of FCR [1]. It is believed that the production of ^101^Tc is optimal when using compact accelerator neutron sources like neutron generators due to its shorter half-life. The use of ^101^Tc would be analogous to the use of short-lived therapeutics and commercially-known positron emitters in the positron emission tomography (PET) imaging modality [19], which require either local or even on-site production in the vicinity of the patient, such as with a mobile imaging unit. We therefore demonstrate the use of neutron generators as a source of neutrons to produce these isotopes combined with an extraction method for removing Tc from bulk Mo. Measurements demonstrating that neutron generators are a potential route to producing these isotopes are presented along with financial considerations for their commercial implementation.

## 2. Results

### 2.1. Production of ^99^Mo, ^101^Mo, ^99m^Tc and ^101^Tc with a Neutron Generator

The liquid ammonium heptamolybdate (AHM) sample was irradiated with neutrons produced by an Adelphi DD110M neutron generator. Despite the high neutron generator yield of 2.2 × 10^10^ n/s the thermal flux measured by gold foil activation was only 1.5 × 10^4^ n/cm^2^·s due to incomplete thermalization. The yield was determined from Bonner ball measurements that were consistent with the known yield from the generator arising from a 160 kV and D+ current of approximately 30 mA. Following irradiation, each sample was transferred to a high purity germanium (HPGe) detector for subsequent counting as shown diagrammatically in Figure 1a in order to detect for the presence of the short-lived species ^101^Mo (t_1/2_ = 14.61 min) and ^101^Tc in the target material. Measurements were carried out for 10 min. The shielded HPGe detector is shown in Figure 1b.

Figure 2 shows the gamma ray spectrum of AHM solution measured immediately following irradiation. Prominent signals from both the shorter-lived isotopes ^101^Mo and ^101^Tc are present directly following irradiation. Characteristic gamma lines for ^101^Mo can be observed at ~590 keV (~19.2%), ~191 keV (18.2%), and 506 keV (11.6%), while gamma lines at ~307 keV (89.4%), ~545 keV (5.96%), and ~531 keV (1.00%) are indicative of ^101^Tc. Much less discernible, but present in the spectrum, are the contributions arising from ^99^Mo at ~740 keV (12.2%) and ~181 keV (6.1%). After the short irradiation and counting time relative to the half-life of ^99^Mo, it would not be expected that significant ^99m^Tc would be present at this point, which is consistent with the data. The yields of ^99^Mo, ^101^Mo, and ^101^Tc in the liquid target at the start of the counting measurement (t = 0) were calculated to be approximately 52 ± 3 Bq (1.3 ± 0.1 nCi), 1.2 ± 0.1 kBq (32.7 ± 3.5 nCi), and 4.6 ± 0.5 kBq (124.1 ± 13.2 nCi), respectively.

Measurements showing the decay of ^99^Mo and subsequent growth of its daughter, ^99m^Tc are shown in Figure 3. The plots were prepared from a series of hour-long spectra. The graphs provide the peak area which was evaluated via numerical integration techniques [20] for the 778 keV line of ^99^Mo and the 140 keV line of ^9m^Tc. The characteristic curve for ^9m^Tc production is a result of the interplay between the half-life of the parent, ^99^Mo (t1/2 = 65.924 h) and that of the daughter, ^99m^Tc (t1/2 = 6.007 h;). It should be noted that ^101^Tc (t1/2 = 14.22 min) and ^101^Mo (t1/2 = 14.61 min;) are also produced, but have much shorter half-lives (Figure 3). Theoretically, the maximum activity of ^99m^Tc obtained from ^99^Mo decay after 22.89 h of growing time [21], which is consistent with the data shown in Figure 3b. The calculated theoretical peak production yield of ^99m^Tc from the decay of 52 Bq of ^99^Mo at approximately this time is approximately 35 Bq (0.95 nCi). 

Additionally, the spectra of ^99^Mo and ^99m^Tc were recorded over a longer period of time (Figure 4). The decay plots were used to determine the decay constants for the 778 keV peak of ^99^Mo and the 141 keV peak for ^99m^Tc. The calculated decay constants for both peaks were approximately 2.75 days or 66 h, which is indicative of the transient equilibrium relationship between the mother-daughter radioisotope pair, where the decay rate is established by the half-life of the mother isotope, i.e., ^99^Mo.

The data shown in Figure 2, Figure 3 and Figure 4 detail the production of ^99m^Tc occurred as a result of irradiation by the neutron generator and also indicates optimal times for extracting the ^99m^Tc from solution.

### 2.2. Isolation of ^101^Tc and ^99m^Tc from Irradiated AHM Solution

Figure 2 shows the presence of both Tc (i.e., ^101^Tc) and Mo (i.e., ^101^Mo and ^99^Mo) isotopes in the irradiated AHM solution present at EOB. The produced Tc, likely in the form of the pertechnetate anion [TcO_4_]^−^, was subsequently extracted from the bulk Mo in the irradiated AHM solution using column chromatography with AC as the column substrate [22]. Gamma ray spectra (Figure 5) of the solution and the extracted sample demonstrate successful removal of ^101^Tc and ^99m^Tc from the solution onto AC. It is noted that isolation of ^101^Tc was near total, however, some peaks of residual ^101^Mo were identified (i.e., 506 keV and 591 keV) in the first hour following extraction (Figure 5a), likely due to inadequate washing of the column. Furthermore, no readily distinguishable peaks above the background from ^99^Mo, which was also present in solution, were identifiable, and the most prevalent species on the AC is ^101^Tc. The extraction performed after one-day was free from any ^99^Mo/^101^Mo, and only ^99m^Tc was found present (Figure 5b).

## 3. Discussion

### 3.1. Production of ^99^Mo/^99m^Tc and ^101^Mo/^101^Tc Using a Neutron Generator and Mo Targetry

The production of the radioisotope mother-daughter pairs ^101^Mo/^101^Tc and ^99^Mo/^99m^Tc using a D-D neutron generator and a liquid Mo target was demonstrated. Whereas the use of a neutron generator for producing ^99^Mo/^99m^Tc from Mo targets for the consideration of commercial application has been previously reported upon [23], it is not apparent that this has been the case for ^101^Mo/^101^Tc. Historically, the ^101^Mo/^101^Tc pair has served as a signature in neutron activation studies, and in the context of ^99^Mo/^99m^Tc it has been considered an impurity coinciding from the use of Mo targets [24,25]. Similarly, ^101^Tc has been reported for cyclotron production of ^99m^Tc, although only as an impurity generated via the ^100^Mo(p, *γ*) reaction. 

Interestingly, the neutron capture properties of ^98^Mo and ^100^Mo are somewhat comparable. For example, the thermal neutron capture cross-sections are 0.130 b and 0.199 b for ^98^Mo and ^100^Mo, respectively, meanwhile the resonance integral cross-sections are 6.70 b for ^98^Mo and 3.76 b for ^100^Mo. Employing fast neutron regimes, such is the case with a D-D generator, permits access through down-scattering interactions, using a light moderator or within the sample itself, into the resonance regions with higher capture probabilities. However, the capture cross-sections for both isotopes drastically fall off at energies above 1 MeV, so it is noted that some moderation is required. Although the targetry investigated here was rudimentary in design, the authors envision the implementation of larger target masses and geometries where the target surrounds the neutron generator at the point of neutron production, better facilitating the interactions previously discussed. 

Another key component of the system is the absence of detection of ^99m^Tc at the time of measurement, i.e., promptly after EOB. As discussed, this results from the short irradiation time relative to the half-life of the parent isotope ^99^Mo, where no significant quantities of ^99m^Tc would have been expected to be produced through the decay of ^99^Mo. It is noted that peak activity of ^99m^Tc as a result of decay typically occurs just short of 23 h-a characteristic feature exploited in the operation of ^99m^Tc commercial generators in radiopharmacies for elution scheduling, where elutions are performed once or twice a day as the industry standard. Therefore, either longer irradiation or longer decay times would generate ^99m^Tc (and ^99g^Tc; t_1/2_ = 211,100 y) in the system. Coincidentally, this phenomenon permits several options when using natural isotopic targets or varying ratios of mixed enriched targets of ^98^Mo/^100^Mo. 1) The isolation of either ^101^Tc or ^99m^Tc, where short irradiations and decay periods are ideal for ^101^Tc, and longer irradiation periods and/or decay periods for ^99m^Tc, or 2) the co-production of ^101^Tc and ^99m^Tc by employing long irradiation times with short decay periods. In the scenario for sole production of ^101^Tc, one drawback using targets containing ^98^Mo would be the potential build-in of ^99^Mo, thus target irradiation schedules would require sufficient decay periods to remove ^99^Mo/^99m^Tc/^99g^Tc from the system, where any residual ^99m^Tc/^99g^Tc could be removed through chemical processing prior to the next irradiation. For the latter production option, the ratio of ^101^Tc:^99m^Tc could be tailored for specific applications through the combination of enrichment ratios of ^98^Mo:^100^Mo and the duration of irradiation. 

Using the extrapolated data, the estimated ^99^Mo (and its daughter ^99m^Tc) production yield as a function of time for a D-D neutron generator outputting 2 × 10^10^ n/s, where a Mo target situated around the generator consumes/captures all of the neutrons produced and contains minimised impurities (an isotopically pure ^98^Mo system is assumed), has been shown in Table 1. It is noted, that for simplicity neutron capture probabilities (i.e., capture cross-sections) are considered negligible, and the number of atoms in the target becomes equivalent to the neutron output (n/s) when complete neutron consumption within the target is assumed. Therefore, using Equation (1), only the neutron output (n/s) and the buildup term [1 − e^−λt^] for ^99^Mo are considered. 

### 3.2. Inventory of Neutron Generators Required for Worldwide Annual Production

With a worldwide consumption of ^99m^Tc in excess of 40,000,000 patient doses annually, where each dose is equivalent to 740 MBq, a significant amount of the parent isotope ^99^Mo must be generated on a weekly basis in order to fulfil demand. This is especially true when factoring in decay following EOB from the reactor or irradiation centre. The use of a neutron generator provides the possibility of generating isotopes at the point of use, for example at a hospital, a temporary field location, oncologist offices, etc. [23]. By using ^99m^Tc from ^99^Mo that is continuously being generated and decaying in real time with a neutron generator, there are less inefficiencies and decay losses due to avoiding unnecessary cooling periods, manipulations, and transportation.

From Table 1, the estimated amount of doses of ^99m^Tc using two different production models was considered. The first model is represented by the commercial industry standard, whereby a target generating ^99^Mo is irradiated for a week, decoupled from the production source, loaded into a generator device, and the ^99m^Tc resulting from the ^99^Mo decay is “milked” approximately every 24 h [26]. The second model assumes the neutron generator is continuously irradiating the target and only the end-product (i.e., ^99m^Tc) is decoupled from the source of production. Under these circumstances, when production in the target material approaches saturation, the amount of ^99^Mo produced is in near steady-state. Thus, the daily amount of ^99m^Tc that can be removed from the system remains constant and does not decrease, as is the case for standard commercial ^99^Mo/^99m^Tc generators. Comparison of the two models and the efficiency gains for continuous generator is shown in Table 2.

As experimentally demonstrated, similar systems could be utilised for producing ^101^Tc, and its application as a direct substitute for ^99m^Tc was considered. For ^101^Tc, the time required to reach peak activity from the decay of ^101^Mo is roughly 21 min. Thus, ^101^Tc in a pure system could be milked approximately every 11 to 22 min from ^101^Mo decay by analogy with standard ^99^Mo/^99^Tc commercial generators. With the persistence of ^101^Tc from ^101^Mo easily upwards of 2 h, a source could be milked 5 to 11 times in this timespan. In comparison to ^99^Mo and ^99m^Tc (half-life ratio ~10.97:1.00, respectively), significant gains in production efficiency for ^101^Tc would be achieved due to the likeness of half-lives of ^101^Mo and ^101^Tc (half-life ratio ~1.03:1.00, respectively). For example, a 22-min period is equal to ~1.51 half-life equivalents of ^101^Mo, whereas a 24-h period is 0.36 half-life equivalents for ^99^Mo. Yields due to daughter decay are also affected, where 22 min is 1.55 half-lives of ^101^Tc, and 24 h is 4.00 half-lives of ^99m^Tc. Therefore, more daughter radioisotope is generated due to faster parent decay and less loss due to daughter decay for ^101^Mo/^101^Tc in a comparable system as the ^99^Mo/^99m^Tc commercial generators. Likewise, approximately 1.2 h would be required for reaching saturation activity (i.e., ~5-half-lives) of ^101^Mo/^101^Tc during irradiation; most commercial irradiations for ^99^Mo/^99^Tc extend up to 7 days or about 50% of saturation activity for ^99^Mo. Furthermore, the decay of ^101^Mo is solely ^101^Tc and the end-decay member of the A = 101 isobar chain is ^101^Ru. Therefore, losses in sequential labelling of ^101^Tc due to lowered specific activity with competition of the presence of other Tc isotopes, such as with ^99m^Tc/^99g^Tc during radiopharmaceutical tagging, is not incurred. It is mentioned that the presence of ^99g^Tc arises from both the decay of ^99^Mo directly (branching decay ~13.95%) and via the isomeric transition decay of ^99m^Tc. 

Shown in Table 3, the total amount of neutron generators required to supply 40,000,000 doses of ^99m^Tc per year (109,589 doses per day) is calculated as a function of doses produced per day using targets of either natural Mo or pure ^98^Mo, and the generator neutron flux, e.g., 2 × 10^10^ n/s. and 2 × 10^12^ n/s, where the former represents the output capabilities of D-D generators, as tested here, and the latter of D-T. The equivalent of ^101^Tc when directly substituted for ^99m^Tc, using a 1:10 ratio factor for ^99m^Tc to ^101^Tc, in a system is also considered.

As a point of comparison, the IAEA reported that there are over 1500 cyclotron facilities distributed around the world [27]. Aiming for this number with the neutron generator model described here, it can be assumed that neutron outputs of ~10^10^ n/s would be too low, even when employing enriched Mo targets, and an excessive number of neutron generators would be required for fulfilling the global daily requirement of ^99m^Tc. However, at neutron outputs of ~10^12^ n/s, meeting this figure is easily achieved. When switching to the ^101^Tc system, even the lower neutron output and natural Mo target provide a comparable number of required generators to the IAEA figure for cyclotrons, whereas at ~10^12^ n/s only a few systems would be required regardless of target composition.

### 3.3. Financial Considerations for Neutron Generator Production

Neutrons can be generated by various means, such as those listed in Table 4. The D-D neutron generators used in this research have both the lowest yield and also lowest cost compared to other approaches presented. Commercially available D-D neutron generators with yields close to 5 × 10^10^ n/s, such as those used in this work [28], have recently become commercially available for applications such as boron neutron capture therapy (BNCT), where they can be deployed as an assembly of multiple generators clustered around a patient [29]. 

Similarly, clusters of D-D generators may be the most cost-effective route to isotope production, especially when production is intended to be deployed close to point-of-use. All of the systems listed in Table 4 require a dedicated facility with sufficient shielding for safe operation. In the case of D-D neutron generators the cost of the facility can be comparable to the cost of the neutron source, primarily because of the relatively low cost of the neutron source. When higher yields are required, linacs, cyclotrons and even reactors may prove the more cost-effective approach, but for small systems, the D-D neutron generator approach may have some merit.

### 3.4. Separation of Tc from LSA Mo Using AC

The affinity and uptake of Tc, particularly as [TcO_4_]^−^, with AC is a well-known phenomenon and has been reported in many studies over the span of several decades [38,39,40,41,42,43,44,45]. Although the mechanism of Tc sorption onto AC is considered a complex process [46], there have been several driving mechanisms identified, such as electrostatic interactions, physisorption, chemisorption/ion-exchange, etc., which have been said to dictate this behaviour [47]. The [TcO_4_]^−^ anion is considered a relatively large anion with a low charge density, a low degree of hydration [48], and exhibits chaotropic character in regards to the Hofmeister series; it is found as the predominant form of Tc under oxidising conditions and across a range of pHs in most non-complexing media. The uptake of Tc on AC occurs at all pHs, however, at acidic pHs below the zero-point-charge of AC, whereby surface species become protonated and the surface becomes positively charged, the kinetics of Tc sorption are significantly enhanced [47,49,50]. Adsorption values greater than 99% [TcO_4_]^−^ (~1 MBq ^99m^Tc) onto AC within 1-min have been reported in the pH range 2–3 where maximum distribution coefficient (*K_d_*) values were measured [51]. The primary contributing surface species are said to be carboxylic, carbonyl, laconic, and phenolic groups, where R–C=O and R–C–OH moieties serve as potential binding sites for [TcO_4_]^−^ [47,49,50] Measured *K_d_* greater than ~10^5^ for Tc uptake onto AC under acidic pHs have been reported [41,47,51]. Even in the presence of competing anions, such as [SO_4_]^2^^−^, [PO_4_]^2^^−^, [F]^−^, [Br]^−^, [Cl]^−^, and [NO_3_]^−^ does this this phenomenon readily occur, although higher concentrations (i.e., [NO_3_]^−^ (> 0.1 mM), [SO_4_]^2−^ and [PO_4_]^2−^ (> 1.0 mM), [Cl]^−^ and [Br]^−^ (> 10 mM)) can be somewhat inhibiting [40,42,49,50]. Due to the lower standard absolute enthalpy of hydration and similar tetrahedral geometry, [ClO_4_]^−^ does effectively outcompete for [TcO_4_]^−^ on AC [49,50]. For comparison, the change in free energy of adsorption of different anions onto activated carbon was determined to be: [ClO_4_]^−^ > [NO_3_]^−^ > [SO_4_]^2−^ > [H_2_PO_4_]^−^ > [ClO_3_]^−^ ≈ [BrO_3_]^−^ > [IO_3_]^−^ [52]. 

The solution chemistry of Mo is much more intricate than that of Tc, where Mo speciation is highly dependent upon pH due to protonation/condensation/oligomerisation reactions, and it is quite sensitive to its chemical environment [53,54]. This behaviour is particularly apparent in acidic pHs where an assortment of anionic polymolydates with varying degrees of hydration and charge densities, and even cationic or neutral species, can (co)exist [53,54,55]. In solution, polyanionic Mo species, such as [MoO_4_]^2−^, are known to form hydrogen bonds stronger than in the surrounding bulk water, classifying them as cosmotropic in nature [56]. It has been shown that Mo adsorption on AC under static conditions is pH dependent, and once again is dictated by the surface charge of AC and the dominant species of Mo in solution [57,58,59,60]. At low pHs (~2–6), whereby the AC surface is positively charged and Mo exists as anionic species in solution (e.g., [Mo_7_O_24_]^6−^, [HMo_7_O_24_]^5−^, [H_2_Mo_7_O_24_]^4−^, and [MoO_4_]^2−^), the interaction of Mo with AC is electrostatically favoured; AC surface functionalities described in the previous paragraph also apply here. Above this region (pH > 6), the AC surface is negatively charged, resulting from the disassociation of weakly acidic sites, and anionic Mo uptake is no longer favoured through electrostatic interactions [57,58]. The *K_d_* coefficients were determined using batch equilibrium techniques for the adsorption of [^99^MoO_4_]^2−^ onto AC with nitric acid concentrations ranging from 0.01 M to 2 M; it was found that *K_d_* values, slightly upwards of ~10^3^, were highest at the lowest concentrations and dropped off rapidly with increasing nitric acid, likely due to the formation of non-anionic species [61]. 

The notion of ^99m^Tc adsorption on AC has been previously discussed in the literature [62]. In fact, the application of using AC to effectively separate ^99m^Tc from LSA ^99^Mo, which was first demonstrated by Tatenuma et al. [63,64,65], has also been the topic of several subsequent studies from other groups [66,67,68]. For example, Tatenuma et al. demonstrated the separation of ^99m^Tc from Mo targets with specific activities <0.5 Ci/g of ^99^Mo at neutral pH; isolation of [^99m^TcO_4_]^−^ in saline solution with a radiochemical purity of 6–7 N within 30–50 min and recoveries between 90–98% were achieved [63]. Production quantities of ^99m^Tc from µCi to hundreds of Cis per batch were determined feasible, and subsequent tagging experiments with HMPAO and MIBI proved comparable to fission-generator-derived ^99m^Tc radiopharmaceuticals in murine models [63].

Presented herein, a similar method of isolating Tc isotopes from irradiated LSA Mo solutions using AC has been demonstrated [22]; however, it is noted that separations were specifically performed under acidic conditions, where the kinetically-driven preferential uptake of Tc as [TcO_4_]^−^ over Mo, likely as anionic polyoxometalate species, onto AC could be exploited. Even at ultra-low specific activities of ^101^Mo/^99^Mo and for the shorter-lived ^101^Tc, effective removal of Tc from the solution was possible using a rudimentary separation platform. 

### 3.5. Technetium-101 as a Potential Ephemeral Diagnostic, Therapeutic, and/or Theranostic Agent

In the field of nuclear medicine, theranostic isotopes are defined as a single isotope or a dual isotope pair, usually of the same element, that exhibits decay characteristics for both therapeutic (i.e., alpha, beta minus, Auger, etc.) and diagnostic decay modes (i.e., beta plus/annihilation, gamma. etc.). This category of isotopes may allow health practitioners to perform radiotherapeutic procedures, for instance for thyroid ablation or in oncological settings where other standard care treatments are not ideal, meanwhile imaging the radiation payload delivered to the site of interest. This may provide practitioners the opportunity to track the progress and success of a procedure as well as ensuring them that unnecessary dose to the patient is not incurred [69]. In contrast to procedures that employ isotopes of homologous elements for the imaging and therapy portions, e.g., ^68^Ga/^177^Lu, the use of single element theranostic systems has the added advantage and assurance that chemical and biochemical behaviour especially in vivo will not differ from each other [70].

The inventory of known technetium isotopes, characterised by masses from A = 85 to A = 120, are radioactive [71]. The hallmark isotope, most commonly used for diagnostic imaging on SPECT systems, for technetium is ^99m^Tc. Although the application of ^99m^Tc in Auger electron therapy has been proposed, other isotopes with more favourable nuclear properties are preferred [72], thus its use as a radiotherapeutic is hardly considered. Likewise, other Tc isotopes ^94m^Tc [73], ^95g^Tc and ^96g^Tc [74] have been identified for diagnostic uses in the PET modality, offering the possibility for acquisition of better resolution data or variation in tracer experiment durations. 

However, it is not readily obvious in the literature that any Tc isotope has been considered or is currently used for therapeutic purposes, outside of the previously mentioned example of Auger therapy for ^99m^Tc. As an alternative, it is typical to substitute either ^186^Re or ^188^Re as a beta-minus emitting radiotherapeutic for ^99m^Tc, being that rhenium is the heavier chemical congener of Tc [75,76]. Although, Re and Tc share many chemical similarities, it is well-known that some pertinent physicochemical properties, such as redox potentials and predominant oxidation states, do differ [77], making the exchange of one for the other not so straightforward [78]. Even in simple systems, such as the binary halides, divergence between the chemistry of the two elements has often been observed [79]. Thus, the availability of a radiotherapeutic/theranostic isotope of Tc would serve as a powerful tool and provide better certainty under these scenarios. Ultimately though, the lack of a previously considered option for Tc likely stems from the fact that most of the accessible beta-minus emitting isotopes of Tc are either considered too long-lived, such as ^98^Tc (t_1/2_ = 4.20 × 10^6^ y) or ^99g^Tc (t_1/2_ = 2.11 × 10^5^ y), or too short-lived, for example ^100^Tc (t_1/2_ = 15.56 s), for this application. 

The general consensus amongst the scientific and medical communities has been that radiotherapeutic isotopes with half-lives ranging in between 6 h to 7 days were usually the most practical and well-suited as discussed by Qaim [80]. Although, Qaim also points out the biological half-life of an isotope is just as important, where consideration of both these parameters biological and physical half-lives constitute effective half-life. Other important factors relate to the pharmacokinetics of the radioisotope, such as the amount of time required for tagging, administration, site accumulation, and site retention, all of which are considered for determining and creating radiation dosimetry models [81]. However, the main argument against shorter-lived radioisotopes manifests from the feasibility and ease of: (1) delivery to the site of use and (2) storage of the isotope/drug.

Expanding interests in therapeutic and theranostic isotopes has presented a host of candidate radioisotopes, including ones that do not necessarily abide by the major criteria outlined by Qaim. A few such examples of these are ^226^Th (t_1/2_ = 30.57 min), which decays by multiple successive alpha emissions to long-lived ^210^Pb [82], and the beta-minus emitter ^214^Pb (t_1/2_ = 27.06 min) and its succession of beta and alpha emitting daughter isotopes [83]. For both radionuclides, integration of the parent isotope, i.e., ^230^U (t_1/2_ = 20.23 d) for ^226^Th and ^222^Rn (t_1/2_ = 3.82 d) for ^214^Pb, into a generator form is a key feature that makes dispensation, tagging, and administering of the end drug possible despite the shorter than usual half-lives [84].

For commercial PET imaging, isotopes with half-lives on the order of minutes to seconds are often implemented, most of which are produced with an accelerator, i.e., cyclotron, for regional distribution and supply, or even directly at the site of use, e.g., hospital. Some of the most notable PET isotopes in this category include ^15^O (t_1/2_ = 122.24 s), ^13^N (t_1/2_ = 9.97 min), and ^11^C (t_1/2_ = 20.36 min). The use of automated synthesis modules and other streamlined manipulation techniques ease the production process, and despite their short-half lives, manufacturing is undertaken in accordance with strict good manufacturing practices (GMP) alongside QA-QC testing prior to dispensation. 

By analogy, the suggested production and use for ^101^Tc is likened to a combination of the aforementioned examples of accelerator-produced, short-lived commercial PET isotopes and the therapeutic radioisotopes ^226^Th and ^214^Pb. Like ^226^Th and ^214^Pb, ^101^Tc is a relatively short-lived isotope and emits fundamental nuclear particles that are within the applicable range for therapeutic and diagnostic procedures. Unlike these isotopes, the parent isotope ^101^Mo of ^101^Tc does not allow for a generator-type scenario of usage. Likewise, this behaviour deviates from the production of PET isotopes, which are typically generated directly via transmutation with the charged particle beam, where no intermediate mother radionuclide precedes the end-product. However, as demonstrated here, ^101^Tc can be produced straightforwardly through neutron capture reactions on ^100^Mo to form ^101^Mo followed by beta-minus decay. The shorter half-lives of both the parent and daughter permit ample activity of ^101^Tc to be produced, even with moderate neutron fluxes and irradiation times under the conditions previously discussed, which in turn would allow for further tagging, QA-QC, and distribution within an immediate locale. Recent efforts on establishing robust and efficient separations of ^99m^Tc from low-specific activity ^99^Mo targets [2,22,23,63,85,86] also provide possible avenues for the isolation and use of ^101^Tc. As demonstrated here, the use of activated carbon as a chromatographic substrate is effective for isolating ^101^Tc from LSA Mo solutions under acidic conditions. When linked to a production source, such as a neutron generator, this particular separation would allow continuous, streamlined and on-demand production and isolation. 

As shown in Table 4, ^101^Tc is a medium energy beta-minus emitter occurring at several prominent energies with the one at ~487 keV (*E_βmax_* = 1320 keV) being the most distinct (~90%). In addition, there are a number of characteristic gamma emissions with the those at 306.8 keV (~89%) and 545 keV (5.9%) being the most notable. The final decay product of the A = 101 isobar chain and daughter product of ^101^Tc is stable 101Ru. This is particularly advantageous considering that no long-lived radioisotopes are generated, such is the case with its sister isotope ^99m^Tc that yields ^99g^Tc, a byproduct often discarded into the environment post-imaging. Although its commercial use has never been proposed, particularly within the medical field, it can be argued that it would have a logical fit within the known toolbox of medical radioisotopes.

In comparison with other current commercially implemented therapeutic/theranostic beta-emitting radioisotopes (Table 5) [87], with the exception of half-life, ^101^Tc exhibits unique, yet consistent decay modes. In fact, the most prominent beta energy at 487 keV is situated between theranostic radioisotopes ^131^I and ^177^Lu on the lower end, and the pure beta-emitters ^89^Sr and ^90^Y on the upper end. Interestingly, ^101^Tc also lies between both radioisotopes of its elemental congener ^186^Re and ^188^Re. However, the primary beta of ^89^Sr is closest in energy of those radioisotopes listed to ^101^Tc, and it has a beta penetration range in tissue of approximately 8 mm [87]. A clinical application of ^89^Sr is targeted radioisotope therapy of osseous metastases; in this regard, it is noted that one of the most common clinical applications of ^99m^Tc diagnostics is for skeletal scintigraphy also known as the “bone scan” with ^99m^Tc-methylene diphosphonate (MDP) or ^99m^Tc-hydroxydiphosphonate (HDP). 

In addition, the primary characteristic gamma decay of ^101^Tc is nearly identical to that of ^131^I at 364.48 keV, which is often used for SPECT imaging. Not only is imaging feasible with commercial SPECT cameras, but ongoing technology, such as Compton cameras also provide an avenue for imaging options of higher energy gammas [17]. It is noted that whole body, simultaneous dual isotopic imaging of ^99m^Tc and ^131^I has been reported for the use of locating functioning bone metastases arising from residual differentiated thyroid carcinoma (DTC) following ^131^I thyroid ablation [88,89]. 

Although the shorter half-life of ^101^Tc may seem to be a disadvantage, there are potential benefits that it may provide. One scenario is by lowering extraneous dose to the patient and public. It is envisioned that multiple therapy sessions could be conducted, with the ability to monitor and adjust the amount of radiation administered, over the same duration observed in a single session for current day radiotherapy treatments with longer-lived isotopes. The shorter half-life would also eliminate the need for patient isolation over extended periods, as such can be the case when high doses of radiation with longer half-lives are administered. Furthermore, from a production and supply perspective, many longer-lived radiotherapeutic isotopes are generated using nuclear reactors with high neutron fluxes in order to achieve workable activities and specific activities [90]. Accommodating for this can be logistically complex from the perspectives of scheduling and distribution, as well as financially expensive. On the other hand, the capacity for sufficient production using accelerators like neutron generators allows many of these issues to be circumvented.

Because of the expansive and exhausted knowledge-base of ^99m^Tc and its utilisation in the field of nuclear medicine [91,92,93], it is not difficult for one to draw conclusions on what the potential possibilities and applications for ^101^Tc could be. Being that it will ideally share nearly identical chemistry and chemical properties as those with ^99m^Tc, most tagging agents and kits would be translatable as well as the biochemical behaviour in vivo of the corresponding tagged moieties. Moreover, opportunities for dual isotopic imaging/therapies with ^101^Tc and any of the aforementioned medically relevant Tc isotopes also becomes plausible. It is not within the scope of this study to pinpoint any particularly one purpose for ^101^Tc as a diagnostic, therapeutic, or theranostic agent, but only to bring light to its prospective use.

## 4. Materials and Methods

### 4.1. Neutron Irradiation

The irradiated sample consisted of a solution of natural AHM (NH_4_)_6_Mo_7_O_24_ (50.00 g, ACS grade, equivalent to 27.15 g Mo) dissolved into ~235 mL of 18 MΩ∙cm^−1^ deionized water (DI H_2_O) acidified (~pH 2) with [HNO_3_] and stored in a polyethylene bottle, as shown in Figure 1b. The sample was placed 6 cm from the neutron source, where it was irradiated for 15 min with an output of 2 × 10^10^ n/s. The neutron source used was a deuterium-deuterium (D-D) neutron generator (a DD-110M, available from Adelphi Technology, Inc. [94]) producing 2.45 MeV fast neutrons. It employed an acceleration voltage of 160 kV and D+ current of approximately 30 mA. The neutron dose and neutron output were confirmed using a Bonner sphere.

### 4.2. Gamma Spectroscopy

An Ortec trans-SPEC-DX-100P N-type HPGe detector was used (Figure 1b) and had been calibrated using NIST traceable calibrated sources. The calibration sources were placed on top of the detector in the same location as the samples to be measured. The HPGe crystal was surrounded by lead shielding to reduce the background signals from the environment. The radioisotope ^40^K is detectable at 1460 keV despite shielding. Using the data from the calibration, the measured efficiencies for the isotopes of interest were determined and are presented in Table 6. 

### 4.3. Measurement of the Neutron Flux (Φ) Based on Gold (Au) Reference

A thin Au reference sample (99.99% purity) with dimensions of 8 mm × 15 mm and a mass of 0.5 g was irradiated for approximately 33.4 min. Activity measurements of the characteristic 411.80 keV peak (95.62%) of ^198^Au (t_1/2_ = 2.964 d) were made 18.5 h after the irradiation and were counted as 261 s. 

When a sample is irradiated with flux, *Φ* thermal neutrons/cm^2^·s, it becomes activated; the activity, in counts per second, *A*, is given by Equation (1):(1)$$A=N\cdot \Phi \cdot \sigma \left[1-e^{\left(-\lambda \cdot t_{\mathrm{irradiation}}\right)}\right]$$
where *N* is the number of atoms in the Au reference sample, *σ* thermal is the neutron absorption cross-section of ^197^Au (98.6 b) and *t_irradiation_* is the irradiation time. The mean lifetime *λ* relates to the half-life, *t*_½_ of the element via Equation (2):(2)$$\lambda =\frac{\log_{e}\left(2\right)}{t_{½}}$$

Therefore, *Φ* can be calculated from the physical measurement of *A* from gamma spectroscopy using Equation (3): (3)$$\Phi =\frac{A}{N\cdot \sigma \left[1-e^{\left(-\lambda \cdot t_{\mathrm{irradiation}}\right)}\right]}$$

The integral of activity (*I*) between the times *t*_1_ and *t*_2_ from the end of the irradiation (*t* = 0) is given by Equation (4):(4)$$I_{0}=\frac{\lambda \cdot Area}{e^{\left(-\lambda \cdot t_{1}\right)}-e^{\left(-\lambda \cdot t_{2}\right)}}$$

### 4.4. Extraction of Tc from Mo in a Bulk Sample of Irradiated AHM Solution

The extraction of Tc isotopes from bulk Mo in an irradiated AHM solution was performed using a column of activated carbon (~1 g). The AHM solution was passed over the column, washed with approximately 1 bed volume of the background solvent, and the column containing sequestered Tc was isolated for subsequent measurement by gamma spectroscopy. 

## 5. Conclusions

In this paper, we demonstrate the production of ^99^Mo/^99m^Tc and ^101^Mo/^101^Tc via neutron bombardment of an aqueous target of natural Mo. Although only relatively small quantities of these particular isotopes were produced under the rudimentary conditions tested, it is believed that improvements in experimental parameters, such as neutron flux, neutron moderation, target mass, target geometry, irradiation duration, extraction conditions, etc., would vastly enhance yields. The quantity of ^99m^Tc and ^101^Tc using neutron generators over an infinite irradiation and isolation period has been extrapolated from this initial irradiation data set. With these data sets, the total number of neutron generators required to meet worldwide annual ^99m^Tc need using a highly mobile, exportable, and non-fission-based system was estimated and the financial implications were presented. Because of the potential efficiency gain in using ^101^Tc for the same or similar imaging modalities and/or theranostics, the wide use of these systems is believed to be advantageous for the purpose of producing ^99m^Tc or ^101^Tc without uranium, a nuclear reactor, or some more expensive, energy intensive, and less mobile accelerator such as a LINAC or cyclotron.

## 6. Patents

The work presented in this manuscript was validation for the patent under review: Mausolf, E. and Johnstone, E. (US/International Patent) Direct, Continuous Transmutation of Molybdenum (Mo) for the Production and Recovery of Technetium (Tc) and Ruthenium (Ru) using a Neutron Source, submitted 2018.

## Figures and Tables

**Figure 1 pharmaceuticals-14-00875-f001:**
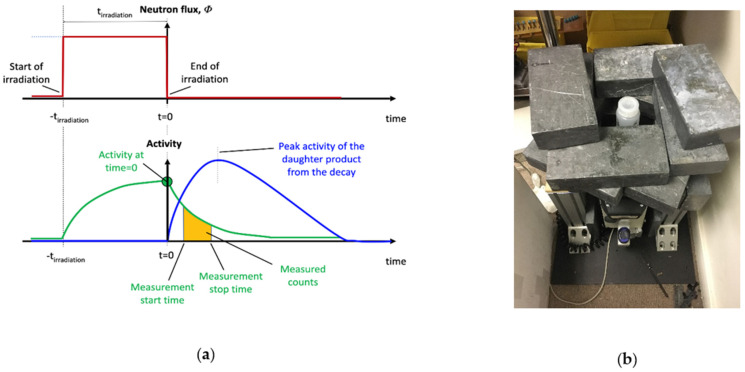
(**a**) Diagrammatic sketch of the irradiation time and isotope production, and (**b**) HPGE detector experimental setup and sample in a polypropylene container.

**Figure 2 pharmaceuticals-14-00875-f002:**
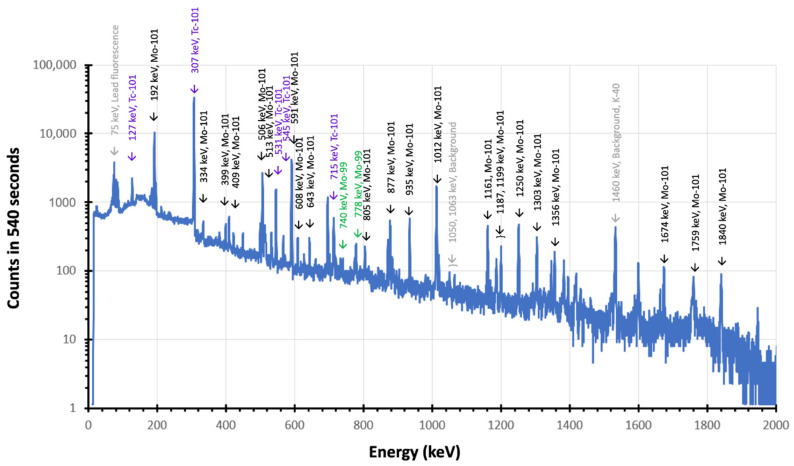
Characteristic peaks for ^99^Mo are shown in green, ^101^Mo in black, and ^101^Tc in purple. Peaks attributed from background contribution are shown in grey.

**Figure 3 pharmaceuticals-14-00875-f003:**
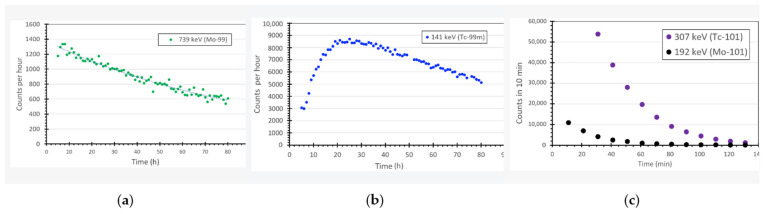
Gamma-ray emission from an irradiated molybdenum-containing sample for which the extraction process was not performed. (**a**) Counts in the 739 keV line from ^99^Mo as a function of time, and (**b**) counts in the 141 keV line of ^99m^Tc as a function of time, (**c**) counts in the 307 keV line for ^101^Tc (purple) and 192 keV for ^101^Mo (black).

**Figure 4 pharmaceuticals-14-00875-f004:**
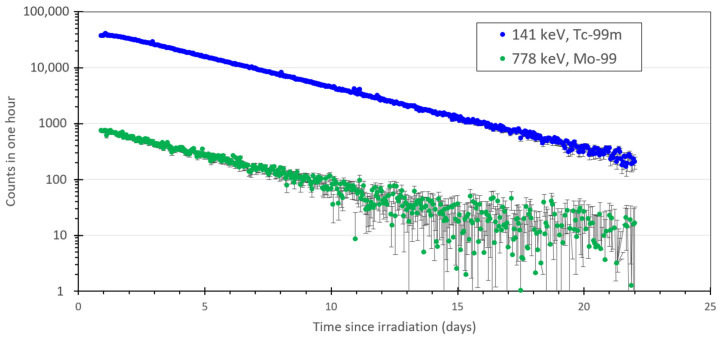
Total counts of ^99^Mo/^99m^Tc following decay of ^101^Mo/^101^Tc from Figure 5 after an 18-h intermediate decay period. The 778 keV line is shown for ^99^Mo (green) and the 141 keV line for ^99m^Tc (blue). Associated error bars are shown in black.

**Figure 5 pharmaceuticals-14-00875-f005:**
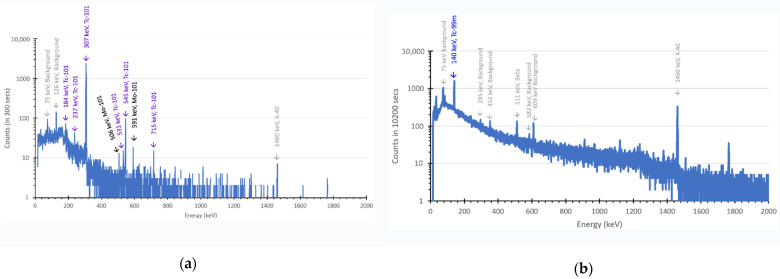
(**a**) Extraction immediately following irradiation yielding ^101^Tc (purple) with residual ^101^Mo (black). (**b**) Extraction after a day of the same irradiated AHM solution yielding only ^99m^Tc (blue). Peaks arising from background contributions are shown in grey.

**Table 1 pharmaceuticals-14-00875-t001:** Production yield of ^99^Mo and ^99m^Tc using a neutron generator outputting 2 × 10^10^ n/s as a function of time. The model assumes complete neutron consumption and transmutation of ^98^Mo to ^99^Mo within the Mo target and concurrent production of ^99m^Tc.

Time (h)	Activity (Bq) Mo-99	Activity (Bq) Tc-99m
1	2.09 × 10^8^	2.02 × 10^7^
24	4.46 × 10^9^	3.70 × 10^9^
168	1.66 × 10^10^	1.47 × 10^10^
330	1.94 × 10^10^	1.72 × 10^10^

**Table 2 pharmaceuticals-14-00875-t002:** Comparison of ^99m^Tc dose output (740 MBq) for a neutron generator (2 × 10^10^ n/s) for the standard commercial ^99^Mo/^99m^Tc generator production model after 1-week of irradiation (*A*_0_ = 16.6 GBq), whereby the target is decoupled from the source of production, versus continuous production at saturation (~5-half-lives; *A_saturation_* = 19.4 GBq), where only the ^99m^Tc is removed from the irradiated target. Assumptions do not account for ^99m^Tc at EOB, generator elution efficiency, nor decay losses due to processing.

Time (Days)	Commercial Generator (Doses)	Continuous Generator (Doses)	Efficiency Gain (%)
1	15	22	44
5	52	110	112
7	60	154	157
14	91	308	238

**Table 3 pharmaceuticals-14-00875-t003:** Estimated production yields and dose requirements associated with ^99m^Tc and ^101^Tc based on 40,000,000 doses per annum (109,589 doses per day) as a function of generator neutron output and Mo target isotopic composition under continuous operation. A ratio factor of 1:10 ^99m^Tc to ^101^Tc was used to estimate doses of ^101^Tc.

**Generator Flux**	2 × 10^10^ n/s				2 × 10^12^ n/s			
**Tc Isotope Produced**	^99m^Tc		^101^Tc		^99m^Tc		^101^Tc	
**Mo Target**	Nat. Mo	^98^Mo	Nat. Mo	^100^Mo	Nat. Mo	^98^Mo	Nat. Mo	^100^Mo
**Doses Generated per day**	5	22	53	220	534	2200	5344	22,000
**Generators Required**	21,918	4981	2068	298	205	50	21	5

**Table 4 pharmaceuticals-14-00875-t004:** Example neutron sources [30,31,32] with estimates of yields and system costs.

Type	Estimate of Beam Energy (MeV)	Approximate Yield Range (n/s)	Approximate System Cost, Order of Magnitude ($M)
Reactor *	Not applicable	>10^17^	~1000
Electron Accelerator ^†^ with Photoneutron Converter	30–40	5 × 10^13^ to 1 × 10^14^	10
Cyclotron ^‡^	10–18.0	5.7 × 10^12^ to 2.1 × 10^14^	1–10
RFQ Linac ^§^	1.5–3.0	1 × 10^11^ to 1.3 × 10^12^	1
D-D Neutron Generator	0.1–0.2	1 × 10^8^ to 1 × 10^11^	0.1–1

* TRIGA reactor $270,000 in 1972 [33], adjusting for inflation is $1.72B in 2021 [34]. ^†^ Financial figures based on 35 MeV, 100 kW electron accelerator described in Ref. [35]. ^‡^ Financial figures based on Ref. [36]. ^§^ Financial figures based on Ref. [37].

**Table 5 pharmaceuticals-14-00875-t005:** Comparison of nuclear data for commercial radioisotopes used for therapeutic/theranostic applications [87] with ^101^Tc.

Isotope	Half-Life	E_gamma_ (keV)	E_beta_ (keV)	Tissue Penetration Range (mm)	Uses
^89^Sr	50.5 d	N/A	587.10 (99.9%)	8	TRT ^1^-osseous metastases
^90^Y	64.00 h	N/A	933.7 (99.9%)	12	TRT-hepatic malignancies, lymphoma
^131^I	8.02 d	284.3 (6.1%), 364.48 (81.5%), 636.98 (7.2%)	191.58 (89.6%), 96.62 (7.2%)	2.4	Diagnostic imaging (SPECT); TRT: thyroid ablation, neuroendocrine tumors, prostate seeds
^177^Lu	6.647 d	208.36 (10.4%)	149.35 (79.4%), 47.66 (11.6%)	2.2	Diagnostic imaging (SPECT); TRT: PRRT ^2^, bone pain palliation, synovectomy, neuroendocrine, metastatic prostate, etc.
^186^Re	3.72 d	137.16 (9.5%)	359.2 (71.0%), 306.1 (21.5%)	4.5	Bone pain palliation, synovectomy, endovascular irrad.
^188^Re	17.01 h	155.04 (15.5%)	795.41 (70.7%), 728.88 (25.8%)	11	Bone pain palliation, synovectomy, endovascular irrad.
^223^Ra *	11.43 d	~82.0 (<2.0%)	492.5 (99.7%), 471.3 (91.3%), 172.9 (0.3%)	-	Bone metastasises
^225^Ac *	9.920 d	218.0 (11.4%), 440.45 (25.9%), 1567.1 (99.7%)	660.34 (97.4%), 492.2 (65.9%), 197.4 (100%), 93.4 (68.8%)	-	Metastatic castration-resistant prostate cancer
^101^Tc	14.22 min	306.8 (89%), 545 (5.9%)	487 (90%), 385 (6%), 127.2 (2.64%)	N/A	N/A

^1^ TRT = targeted radionuclide therapy; ^2^ PRRT = peptide receptor radionuclide therapy. * Primarily ⍺-decay, yet exhibit multiple decay modes, i.e., β, and γ, and daughter products; for comparison only β and γ reported from decay chain

**Table 6 pharmaceuticals-14-00875-t006:** Measured isotopes and emission lines of interest with corresponding detector efficiencies.

Isotope	Energy of Key Line (keV)	Half-Life	Detector Efficiency (%)
^101^Mo	192	14.61 m	9.4 ± 1
^101^Tc	307	14.22 m	6.6 ± 0.7
^99^Mo	778	65.924 h	3.3 ± 0.3
^99m^Tc	141	6.001 h	11.8 ± 1.5
^198^Au	411	2.697 d	5.4 ± 0.5

## Data Availability

Data is contained within the article.

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
