# Peer review of "Fusion-Based Neutron Generator Production of Tc-99m and Tc-101: A Prospective Avenue to Technetium Theranostics"

_pharmaceuticals, 2021, doi:10.3390/ph14090875_

Round 1

Reviewer 1 Report

Review: Fusion-based Neutron Generator Production of Tc-99m and Tc- 2 101: An Avenue to Technetium Theranostics.

Mausolf, E., et al Pharmaceuticals.

The manuscript describes some interesting concepts relating to the preparation of 99mTc and 101Tc from their respective 99Mo and 101Mo sourced from neutron generator. The authors have presented some very good and objective points regarding the current production and availability of 99Mo- 99mTc, as well as its inefficiencies and constraints. Adding to this are the constant threats of the supply chain (as discussed on page 3) which have at times caused serious disruptions to the availability of 99mTc for nuclear medicine. Overall, most of these points are valid and as such the concept of using neutron generators to produce radionuclides is an attractive option and is similar to that used by cyclotrons to produce PET radionuclides. The second point of discussion in this manuscript deals with the preparation of 101Mo and 101Tc, the latter of which, a b-emitter has been proposed as a therapeutic radionuclide. Irrespective of the feasibility the concepts presented would be of interest to many readers in the field.

The authors raised several key points regarding the true costs of 99Mo production using reactors as the costs of construction, operations, waste management etc., are not necessarily covered by nominal irradiation charges. From this perspective technological improvements, alternative methodologies such as these described in this manuscript are certainly warranted and needed. Furthermore, the establishment of small medical cyclotrons, their operations and product delivery to the market seem to represent a much clearer cost than the 99Mo/99mTc scenario as many private organisations have invested in cyclotron technologies for radionuclide and radiopharmaceutical production whereas no private organisation has invested in a nuclear reactor. The concept of using low cost neutron generators would not only be of interest for these radionuclides, but for others as well.

However, some of the points presented are also speculative which the authors need to consider or modify.

  1. Comment: Although 99mTc is an essential and widely used radionuclide for nuclear medicine, does it still constitute 85% of the global radiopharmaceutical market. This figure was correct at its peak about 20 years ago, but since then we have seen double digit growth in PET and a marked decline in SPECT. Does this number need revision? Recent OECD/NEA estimates predicted a 25% decline in the demand for 99Mo globally. As the authors have correctly pointed out this has been exacerbated by disruptions in supply and shifting imaging protocols away from SPECT to PET. This trend is being further enhanced by the expansion of PET/MRI and more than likely when whole body PET-CT is more readily available.
  2. In the introduction on line 43 the authors describe the current production process for 99mTc from 99Mo as a ‘fundamentally flawed design’. Perhaps the word ‘flawed design’ is a bit too harsh considering that the advent of the whole 99Mo-99mTc process, its use and adoption by the nuclear medicine market was never designed- rather the technology just evolved during the last 50-60 years. Sure, it has constraints and many inefficiencies, it is heavily subsidised, and its price does not include the construction costs of the reactor nor waste management. However, because of its convenience and availability, it has become the most widely used radionuclide in nuclear medicine. For an industry with 40,000,000 doses annually (Authors figures) I would consider it a success and not flawed. Of course, all technologies need constant revision, updates or replacement.
  3. In lines 47-48 the authors question and describe the radiopharmaceutical industry’s losses and feasibility compared to other industries. In principle this is correct where <10 % of product reaches its end user- the patient. However, this argument is somewhat simplistic as the same can be said about any radionuclide e.g., F-18 for FDG, carbon-11 or 68Ga for PSMA. The same can be said about the 68Ge/68Ga generator, as the parent isotope is continuously decaying 24 h per day for the life of the generator over 6-9 months with perhaps 2-3 elutions per day excluding weekends and holidays. Yet the 68Ga-radionuclide, products are in high demand and provide enormous benefit to patients with real costs recovery. Yes, in principle because of decay; it is a very inefficient process, but immensely useful in medicine. The fundamental decay process of all radionuclides will therefore be a constraint in this industry irrespective of the radionuclide including 101Mo/101Tc.
  4. The concept and choice of 99mMo production via fission vs 98Mo using (n, g) has been around for several decades, yet the fission process has been ultimately used from a logistics / economic perspective to meet global demand and to get around the 98/99Mo specific activity issue when loading onto the generator. How do the authors propose to work around the similar problem with the neutron generator?
  5. 99mTc shortages have stimulated considerable, alternative production methods using cyclotron technologies via the 100Mo(p,2n)99mTc. Have the authors made a comparative analysis of these developing technologies with those proposed?
  6. Lead-up times and availability of neutron generators, costs etc. The authors gave a detailed analysis of the irradiation parameters that could produce batches of 99Mo/99mTc and 101Mo/101Tc, at two different neutron fluxes as well as extrapolating out to the number of generators required to produce various dose quantities. However, no explanation or comment has been made on the feasibility of this technology to achieve these and on what time scale. How would the costs compare say to cyclotrons, set-up, operations etc? Perhaps the authors could provide a brie insight into this.
  7. The authors gave a detailed proposal on the potential use of 101Tc as a dual radiotherapeutic / imaging agent. Although 101Tc has a medium b-energy at ~487 keV (Eβmax = 1320 keV, 90%) and is in between that of I-131 and Lu-117, its half-life is only 14.22 minutes which would be difficult to justify as a therapeutic both in terms of its handling, radiopharmaceutical preparation, QC and biological use. Considering the large numbers of b- and a-emitting radionuclides under development, it would be difficult to convince readers of its potential in the absence any data and logistics.

  1. Although, the use of Tc-99m will continue to be used, over the next few decades, there is less interest in Tc-radiopharmaceuticals and SPECT as an imaging agent. Consequently, there has been little development and subsequent approval of 99mTc radiopharmaceuticals.

Reviewer 2 Report

This manuscript describes an alternative production scheme for Tc-99m and Tc-101 employing d-d neutron generator.  The topic itself is quite interesting. However, in overall the quality of the paper needs to be significantly improved to be publishable in Pharmaceuticals.

I am missing here clear comparison of various methods for Tc-99m production, particularly the reactor based fission methods and cyclotron production methods.  

Further, beside the comparison of production methods, the distribution channels/strategies should be evaluated and clarified.  Detailed  comparison and evaluation is missing here. 

Schemes 1-3 does not bring any new and significant information. 

The amount of experimental work is quite low. It could be taken as a proof of principle if some additional experimental details are given. Total overall neutron flux of 10^10 is quite low to produce significant amounts of Mo/Tc under reasonable conditions. Please comment in more detail. Enormous amount of expensive enriched target material is needed to produce relevant amounts (of low specific activity) Mo. Also the radionuclidic purity of the final product required by FDA/EMA is questionable. Please comment in more detail on practical feasibility of your method. 

Delete Table 1 - instead of this, please give clear yield of Mo-99 and Tc-99m for 1 sec. pulse and calculate the production yield of all isotopes for e.g. one week irradiation, taking into account consecutive equilibrium between Mo/Tc and their partial decay during irradiation/processing. 

Table 2. contains some numerical error. You will never get 251,4 doses of Tc-99m (740 MBq) in one week using flux of 10^10 neutrons/s. In Table 1 you mentioned that you produce 14 kBq of Mo-99m in 1 sec. Neglecting its decay you will get something like 8,5 GBq of Mo-99 in 7 days that approximately corresponds to one Mo/Tc generator used in nuclear pharmacies. How you intend to process such low specific activity Mo-99 to receive Tc-99m? It will also take some time (and decay of Mo-99/Tc-99m).

Figure 6 is redundant. 

Table 4 should contain also some alpha emitters like Ac-225 or Ra-223.

Author Response

This manuscript describes an alternative production scheme for Tc-99m and Tc-101 employing d-d neutron generator.  The topic itself is quite interesting. However, in overall the quality of the paper needs to be significantly improved to be publishable in Pharmaceuticals.

  1. I am missing here clear comparison of various methods for Tc-99m production, particularly the reactor based fission methods and cyclotron production methods.  

The primary interest of the authors was for the comparison of neutron-driven production routes, which is primarily comprised of (n, gamma) reactions on either fission-based targets or Mo-based ones. Because PET and SPECT production modes are often thought as two completely different logistical dogmas (the former utilising a cyclotron and the latter most typically a reactor), the likeness of cyclotron production of short-lived PET isotopes was only exercised in order to make a comparison to the production of SPECT ones using a compact accelerator neutron source, such as a neutron generator, where regional production and supply would occur. It is true that the distribution model of the proposed neutron generator scheme of production would be nearly identical to that of cyclotron production, although the production modes themselves and the physical and nuclear behaviours they are contingent upon have very little in common. It was not the intent of the manuscript to compare the varying modes of production as this has been the topic of previous publications, for example the one from Wolterbeek et. al [14]. This reference has been added into the introduction after the statement, “There are several production routes for generating 99mTc…”

  1. Further, beside the comparison of production methods, the distribution channels/strategies should be evaluated and clarified. Detailed comparison and evaluation is missing here. 

The current reactor-based 99Mo / 99mTc supply chain has been intricately detailed in the OECD reported from 2019 cited in the manuscript. Because of the intimate relationship of many PET imaging agents with current radiotherapeutic / theranostic procedures, the authors assume that the reader base is familiar with the supply chain of these isotopes, specifically those related to cyclotron production. The analogy was made between the neutron generator and the PET models, because the authors believe the two would be very similar.

  1. Schemes 1-3 does not bring any new and significant information. 

The authors have removed Schemes 1–3 from the manuscript.

  1. The amount of experimental work is quite low. It could be taken as a proof of principle if some additional experimental details are given. Total overall neutron flux of 10^10 is quite low to produce significant amounts of Mo/Tc under reasonable conditions. Please comment in more detail. Enormous amount of expensive enriched target material is needed to produce relevant amounts (of low specific activity) Mo. Also the radionuclidic purity of the final product required by FDA/EMA is questionable. Please comment in more detail on practical feasibility of your method. 

The total overall neutron flux reported herein is admittedly much lower than typical industry standards, but as the Reviewer has pointed out, comparatively much larger target masses, and stated in the manuscript, a multiplicity of neutron generators and longer irradiation durations at this output will be required for compensation. It is noted that the IAEA has reported that there are over 1500 cyclotron facilities throughout the world [15], thus by analogy, it is not absurd to think that a distributed model of compact accelerator neutron sources that minimise decay losses could approach the current requirement needs.

For example, Leung et. al. calculated the production of 99Mo through the (n,γ) and (n,2n) using a high-yield accelerator source, i.e., Li-D or D-T, with an output approaching 1012 n/mA·s. It was determined that in a concentrated target volume, i.e, 6.2 x 1022 atoms/ cm3, placed 1 cm from a source with a flux of ~1010 n/cm2·s over 100 h that ~124 mCi of 99Mo could be produced with resonance neutrons, or ~270 mCi could be generated with fast neutrons via the 100Mo(n,2n) reaction under similar conditions [16]. Although these conditions are somewhat different than those reported in the manuscript, it clearly demonstrates the feasibility of the proposed model.

For radionuclidic purity required by FDA/EMA, it can be argued that it is more of an issue for other modes of production, e.g., using a cyclotron for direct production of 99mTc, whereby other longer lived-isotopes of Tc are generated via transmutation / decay concurrently with 99mTc and cannot be separated from the final product [17]. As described in the manuscript, this would not be the case under the proposed scenarios.

  1. Delete Table 1 - instead of this, please give clear yield of Mo-99 and Tc-99m for 1 sec. pulse and calculate the production yield of all isotopes for e.g. one week irradiation, taking into account consecutive equilibrium between Mo/Tc and their partial decay during irradiation/processing. 

The authors have not deleted Table 1, but instead they have modified it and along with Table 2 to include the information that the Reviewer has requested. This is further addressed in the subsequent point below.

  1. Table 2. contains some numerical error. You will never get 251,4 doses of Tc-99m (740 MBq) in one week using flux of 10^10 neutrons/s. In Table 1 you mentioned that you produce 14 kBq of Mo-99m in 1 sec. Neglecting its decay you will get something like 8,5 GBq of Mo-99 in 7 days that approximately corresponds to one Mo/Tc generator used in nuclear pharmacies. How you intend to process such low specific activity Mo-99 to receive Tc-99m? It will also take some time (and decay of Mo-99/Tc-99m).

The authors acknowledge that the the Reviewer is correct. There was an error in the calculation - it is not possible to get ~250 doses of 99mTc using a flux of ~2 x 1010 n/s even in a scenario of complete neutron consumption and transmutation of 98Mo + n → 99Mo. For an irradiation period of 168 hours, the calculated production would be approximately 16.6 GBq (448 mCi) of 99Mo and 14.7 GBq (398 mCi) of 99mTc or 20 doses at EOB. If the activity at EOB generator elution efficiency are disregarded, then a generator starting with ~16.6 GBq of 99Mo (assuming no loss due to processing) would produce approximately 52 doses over a 5-day workweek.

However, in the scenario proposed by the authors, if the 98Mo is continuously irradiated and production approaches the limit of saturation, say after 330 hours (~5-half-lives), the yields are slightly more with approximately 524 mCi and 465 mCi of 99Mo and 99mTc, respectively, or 22 doses at EOB. If only the 99mTc is removed and production of 99Mo persists with an activity of approximately ~524 mCi, then every 24 hours approximately 439 mCi of 99mTc would be generated, or ~22 doses/day, which would correlate to ~110 doses for a 5-day workweek. Compared to the typical generator scenario, where the parent isotope is not being continuously regenerated, the efficiency gains in doses is approximately 112% for a 5-day workweek. For a full 7-day week and a 2-week period the gains in efficiency increase to 157% and 238%, respectively.

There have been numerous publications on methods for making low-specific activity 99Mo viable [18, 19]. The authors are not convinced that they all would effectively translate into the commercial setting, and because of this, the authors have provided a select few possibilities in the manuscript that they believe would work. That being stated, there have been several routes that have been implemented in the past in a  commercial setting, i.e., gel-generators, liquid-liquid extraction with methyl ethyl ketone (MEK), and thermochromatography, and of these several, as well as a few others, e.g., aqueous biphasic extraction chromatography (ABEC), are still being championed as solutions today.  However, the authors have included some experimental data showing Tc removal from a LSA Mo target using activated carbon as a solid substrate for sequestration - this data method has been added into the methods along with reasoning / supportive literature behind its application in the discussion.

  1. Figure 6 is redundant. 

The authors have removed Figure 6 from the manuscript.

  1. Table 4 should contain also some alpha emitters like Ac-225 or Ra-223.

The authors agree with the Reviewer and have added these into the table. Although these primarily decay via alpha, which was not the real intention for comparison, it is acknowledged that they have several beta decays associated with them as well. Especially considering the application of 223Ra with bone metastasises and the implications of Tc for bone imaging pointed out in the paper, this makes a nice parallel.

Reviewer 3 Report

This paper does not comply with 2 major requirements for publication:

1) Scientific. Used methodologies are not clearly described, ensuring scientific reproducibility by other authors.
The very few lines on the experimental methods, lines 418 to 430 (in clear discrepancy with a long 12 pages introduction) are vague and do not properly describe the experimental setup (there is a reference to a fig 1, in line 424, that is not a experimental setup).

Results are presented without statistical discussion or uncertainties.

The large introduction is argumentative (nearly speculative) about a "economic" merit of the process to be presented, but does not really quantify.

Furthermore, it ignores the last decade scientific papers concerning production of Tc99m in cyclotrons (that resulted in available commercial products) compromising a quality state of the art

 2) Compliance with the scientific scope of the journal: it refers to a production method for Tc99m from the physics/nuclear science perspective, not evidencing the properties of the product as a (radio)pharmaceutical.

As the poor experimental description is understandable in the context of some patents pending a re-submission, eventually after patent process, is recommended as the paper in the present status is not compatible with the scientific quality of this journal.

Author Response

This paper does not comply with 2 major requirements for publication:

  1. (1) Scientific. Used methodologies are not clearly described, ensuring scientific reproducibility by other authors.

The authors will comment that multiple experiments have been performed surrounding the fundamental  concept of the manuscript: production of 99mTc and 101Tc from a Mo target using a neutron generator as a source of neutrons. These species have been reported in the literature during the usage of higher output neutron sources, which have been alluded to and referenced in the manuscript, and have been unequivocally  observed in the presented data. Further comments on this subject are found in the following point.

  1. The very few lines on the experimental methods, lines 418 to 430 (in clear discrepancy with a long 12 pages introduction) are vague and do not properly describe the experimental setup (there is a reference to a fig 1, in line 424, that is not a experimental setup).

The MDPI publication format for Pharmaceuticals is arranged in so that the experimental work “Methods and Materials" is found located after the “Conclusion” of the manuscript. It is commented that this is not a feature unique to the Pharmaceuticals journal alone. In addition, the Introduction is composed of 6 paragraphs allotted over approximately 3 pages, which included 3 figures (now removed), not 12 pages. The authors do concede that the “Discussion" section is longer, however, it was assumed that taking the liberty to discuss a topic, especially one that has never been presented in prior literature, may require this. On the contrary, the “Materials and Methods" section is concise and effective. Admittedly, the experiment itself is quite simplistic, which in actuality should make it easily reproducible.

The authors do agree with the Reviewer that the reference to Figure 1 was not correctly referring to an experimental setup for irradiation - it is the sample being measured on the gamma spectrometer. The intention of the authors was to refer to the sample itself in the polyethylene bottle, which is shown in Figure 1, when it was described. This was simply a grammatical mistake. Thus, the excerpt:

 “…stored in a polyethylene bottle. As shown in Figure 1, the sample was placed 6 cm…”

should actually read:

“…stored in a polyethylene bottle, as shown in Figure 1. The sample was placed 6 cm…”

This excerpt has been updated in the manuscript.

  1. Results are presented without statistical discussion or uncertainties.

The authors agree with the Reviewer. Statistical discussion / uncertainties have been assigned where appropriate.

  1. The large introduction is argumentative (nearly speculative) about a "economic" merit of the process to be presented, but does not really quantify.

The authors have cited the 2019 OECD report on 99Mo / 99mTc production and distribution, which is the most recent of several OECD reports that has covered this topic and details the economics of the industry. From the report:

“This lack of investment has resulted in a system reliant on older reactors that have had reliability

concerns over the last decade. The shortage seen in 2009 and 2010 is a symptom of this economic

problem. Once the shutdown reactors return to operation and the short-term supply becomes stable

again, it is important to stress that although the symptom has been addressed, the underlying problem

– the unsustainable economic structure – has not.”

Fundamentally, because the current system fails to achieve a full cost recovery (FCR) model, even when it does not account for the financial aspects of processing, logistics, security, storage, etc. that are required of frontend uranium enrichment and backend fission waste disposal (either for the targets themselves or the fuel in the reactors), nearly any other model that can meet the supply and demand of the market that does not require these features is arguably better. In this regard, the concept of Mo-based targetry and a fusion-based neutron generator for the production of these isotopes as well as 101Mo / 101Tc without the use of fissionable materials does have merit.

However, as shown in Point #6, the authors have added in a section into the manuscript on financial considerations for production in comparison to other known neutron sources.

  1. Furthermore, it ignores the last decade scientific papers concerning production of Tc-99m in cyclotrons (that resulted in available commercial products) compromising a quality state of the art

This concern has been priorly addressed in Points #5 and #9 - the authors would kindly request the Reviewer to refer to these.

  1. (2) Compliance with the scientific scope of the journal: it refers to a production method for Tc-99m from the physics/nuclear science perspective, not evidencing the properties of the product as a (radio)pharmaceutical.

As the poor experimental description is understandable in the context of some patents pending a re-submission, eventually after patent process, is recommended as the paper in the present status is not compatible with the scientific quality of this journal.

This concern has been priorly addressed in Point #7 - the authors would kindly request the Reviewer to refer to it.

Additionally, for the special issue entitled, "Design, Synthesis and Biological Testing of Next Generation Theranostic Radiopharmaceuticals,” for which this manuscript has been submitted, the description of intention includes:

“However, the rising availability of suitable therapeutic radionuclides, and the emergence of new targets and radiochemical strategies, are making the development of new theranostic platforms increasingly relevant. With this, researchers are invited to submit original and review articles of a basic, preclinical and clinical scope, focusing on the design, synthesis and biological testing of new radiotheranostic compounds and delivery systems, and their future potential. Work that only focuses on diagnostics or therapy should have a clear potential application in theranostics.”

The authors that believe the presented work abides by these conditions. Even if the discussion of 101Tc as a  potential theranostic itself is disregarded, the role of 99mTc in both the diagnostic and therapeutic  / theranostic modalities is unquestionable, and novel modes of its production should be relevant and of interest to the community. The objective behind the argument for 101Tc as a potential Tc theranostic isotope is to stimulate new discussion in this exciting and rapidly developing field, where different modes of production and distribution that open the possibility for novel therapeutic isotopes to be used should be considered.

In order to permit any confusion behind this, the title of the manuscript has been changed to, “Fusion-based Neutron Generator Production of Tc-99m and Tc-101: A Prospective Avenue to Technetium Theranostics.”

References

  1. Lee D. S.; Lee, Y. S.; Lee J. S.; Suh, M.S. Promotion of Nuclear Medicine-Related Sciences in Developing Countries. Nucl Med Mol Imaging. 2019, 53(2), 73– doi:10.1007/s13139-019-00583-0
  2. https://www.cms.gov/medicare-coverage-database/details/nca-decision-memo.aspx?NCAId=279&type=Closed&bc=AiAAAAAACAAAAAAA&
  3. Wondergem, M.; van der Zant, F. ; Knol, R. J. J. et al. 99mTc-HDP bone scintigraphy and 18F-sodium fluoride PET/CT in primary staging of patients with prostate cancer. World J Urol., 2018,36, 27–34. https://doi.org/10.1007/s00345-017-2096-3
  4. Ljungberg, M.; Pretorius, P. H. SPECT/CT: an update on technological developments and clinical applications. Br J Radiol. 2018, 91(1081), doi:10.1259/bjr.20160402
  5. https://www.beloitdailynews.com/news/local-news/two-electron-beam-accelerators-delivered-to-northstar/article_79a4dfdb-e9d2-54e4-a7cf-0ad925d0b04c.html
  6. Ruth, T. J. The Shortage of Technetium-99m and Possible Solutions. Ann. Rev. of Nucl. Part. Sci.202070:1, 77–94.
  7. Abdel-Dayem, H. M. Congenital Absence of Submaxillary Gland Detected on 99mTc-Pertechnetate Thyroid Imaging. Nucl. Med. 1978, 3, 442.
  8. Kiratli, P.O., Aksoy, T., Bozkurt, M.F. et al. Detection of ectopic gastric mucosa using 99mTc pertechnetate: review of the literature. Ann Nucl Med., 200923, 97–105. https://doi.org/10.1007/s12149-008-0204-6
  9. (a) Lemb, M., Oei, T.H., Eifert, H.et al. Technegas: a study of particle structure, size and distribution. Eur J Nucl Med., 199320, 576–579. https://doi.org/10.1007/BF00176550; (b) Lloyd, J.,; James, J.; Shields, R. et al. The influence of inhalation technique on Technegas particle deposition and image appearance in normal volunteers. Eur J Nucl Med. 1994, 21, 394–398. https://doi.org/10.1007/BF00171413
  10. Vogel, W. V.; van der Marck, S. C.; Versleijen, M. W. J. Challenges and future options for the production of lutetium-177. J. Nucl. Med. Mol. Imaging2021. https://doi.org/10.1007/s00259-021-05392-2
  11. Engle, J. W. The Production of Ac-225. Radiopharm. 2018,11(3). https://doi.org/10.2174/1874471011666180418141357
  12. Alberto, R. The “Carbonyl Story” and Beyond; Experiences, Lessons and Implications. ChemBioChem202021, 2743.
  13. https://www.annualreports.com/HostedData/AnnualReports/PDF/NASDAQ_LNTH_2020.pdf
  14. Wolterbeek, B.; Kloosterman, J.L.; Lathouwers, D.et al. What is wise in the production of 99Mo? A comparison of eight possible production routes. J Radioanal Nucl Chem., 2014302, 773–779. https://doi.org/10.1007/s10967-014-3188-9
  15. https://www.iaea.org/newscenter/news/cyclotrons-what-are-they-and-where-can-you-find-them
  16. Leung, K.; Leung, J. K.; Melville, G. Feasibility Study on Medical Isotope Production using a Compact Neutron Generator. Rad. Isotop., 2018, 137, 23–27.
  17. Melendez-Alafort, L.; Ferro-Flores, G.; De Nardo, L.; Bello, M.; Paiusco, M.; Negri, A.; Zorz, A.; Uzunov, N.; Esposito, J.; Rosato, A. Internal radiation dose assessment of radiopharmaceuticals prepared with cyclotron-produced 99m Medic. Phys. 2019, 46, 1437– 1446.
  18. Dash, A.; Knapp Jr., F. F.; Pillai M. R. A. 99Mo / 99mTc separation: An assessment of technology options. Med. Bio. 2013, 40, 167–176. 17
  19. Gumiela, M. Cyclotron production of 99mTc: Comparison of known separation technologies for isolation of 99mTc from molybdenum targets. Nucl Med Biol. 2018, 58, 33-41. doi: 10.1016/j.nucmedbio.2017.11.001.

Round 2

Reviewer 1 Report

The modifications made by the author are appropriate and the paper reads very well.

No further changes required.

Reviewer 3 Report

The authors must be congratulated for the incisive addressing of all raised points and for significant improvement of the paper, now much more clear, objective and scientific oriented.
Proposed title is now more adequate. Bibliographic references revision also improved the quality of this section.